# Ultrasonographic Assessment for Tenosynovitis in Juvenile Idiopathic Arthritis with Ankle Involvement: Diagnostic and Therapeutic Significance

**DOI:** 10.3390/children9040509

**Published:** 2022-04-03

**Authors:** Sara Della Paolera, Serena Pastore, Alen Zabotti, Alberto Tommasini, Andrea Taddio

**Affiliations:** 1Institute of Maternal and Child Health IRCCS “Burlo Garofolo”, Via dell’Istria *n*° 65/1, 34137 Trieste, Italy; saradellapaolera@gmail.com (S.D.P.); alberto.tommasini@burlo.trieste.it (A.T.); andrea.taddio@burlo.trieste.it (A.T.); 2Department of Medical and Biological Sciences, Rheumatology Clinic, University of Udine, Piazzale Santa Maria della Misericordia *n*°15, 33100 Udine, Italy; zabottialen@gmail.com; 3Department of Medicine, Surgery and Science Health, University of Trieste, Piazzale Europa *n*°1, 34127 Trieste, Italy

**Keywords:** ankle, juvenile idiopathic arthritis, ultrasonography, methotrexate, biological therapy

## Abstract

Background: The role of musculoskeletal ultrasound in JIA is still controversial, although there is growing evidence on its utility, especially in the diagnosis of tenosynovitis. Methods: We presented a retrospective cross-sectional study of a group of patients with JIA with ankle swelling followed in a Pediatric Rheumatology Service of a tertiary-level pediatric hospital in Northern Italy during the follow-up period between January 1st 2003 and December 31st 2019. Preliminary results have been presented at the EULAR Congress 2021. We enrolled only patients who underwent msk-US, and we identified those with a clinical and sonographic diagnosis of tenosynovitis. For each patient, we collected data on demographics, clinical characteristics, and therapeutic strategies during the follow-up. Results: On December 31st 2019, 56 swollen ankles of 48 patients were assessed with msk-US. Twenty-two ankles showed sonographic signs of joint synovitis, sixteen ankles presented signs of both joint synovitis and tenosynovitis, and fourteen ankles presented sonographic signs of tenosynovitis only. Overall, tenosynovitis was detected on 27 (56%) out of 48 children with at least a swollen ankle. In 13 patients out of 27 with tenosynovitis (48%), there was no joint synovitis of ankle or foot. Twenty-five patients with tenosynovitis (92%) achieved clinical and radiological remission: seven patients achieved remission of tenosynovitis with methotrexate only, and fifteen patients with biological drugs alone or in combination therapy. Conclusions: We observed that more than half of the patients with ankle swelling presented a tenosynovitis, and about 50% of them did not show sonographic signs of an active joint synovitis. Among patients with tenosynovitis, biological therapy alone or in association with DMARDs showed effectiveness in inducing disease remission.

## 1. Introduction

Tenosynovitis is one of the manifestations of Juvenile Idiopathic Arthritis (JIA), and may be suspected in the presence of swelling, pain, or tenderness along the length of tendons [1]. Tenosynovitis may be clinically difficult to distinguish from synovitis, especially in young children with ankle involvement [2]. The role of musculoskeletal ultrasound (msk-US) in the management of JIA is still controversial [3]. The use of msk-US in JIA diagnosis and follow-up is still discussed, as well as its role in helping the physician in therapeutic choices. However, msk-US seems to be very useful in differential diagnosis between joint synovitis, tenosynovitis, or soft tissue edema [3,4].

Tendon inflammation is often associated with synovitis of one or more adjacent joints, and it has been reported in up to 71% of patients with ankle arthritis. In these cases, the clinical examination alone can be misleading and not sufficient to adequately recognize and treat tenosynovitis [4]. 

To the best of our knowledge, there is limited literature regarding diagnosis, clinical management, and therapeutic options for tenosynovitis in children, except for the use of glucocorticoids tendon sheath injections (CS-TSI) [1,5]. 

The first aim of our study was to define the prevalence of tenosynovitis of the ankle in patients with JIA and ankle swelling.

The secondary aim was to describe the clinical characteristics of patients with tenosynovitis, and to analyze the response to treatments.

Preliminary results of the study have been presented at the EULAR Congress 2021 [6].

## 2. Materials and Methods

### 2.1. Patients

We conducted a retrospective cross-sectional study among patients with JIA cared for at the Rheumatology Service of the Institute for Maternal and Child Health IRCCS “Burlo Garofolo” of Trieste, Italy. Diagnosis of JIA was made according to the ILAR criteria [5], and all patients were aged less than 18 years at time of recruitment. 

We enrolled all of the patients who reported a swelling of the ankle at least once during a follow-up period between 1 January 2003 and 31 December 2019. Among these patients, we included only those who underwent msk-US. Based on both clinical and sonographic examination, we identified patients with tenosynovitis, and we described their demographic and clinical characteristics, as well as the therapeutic approach undertaken. 

Since there are no validated scores to define disease remission in tenosynovitis, we considered remission the absence of clinical and radiological signs of tendon inflammation for at least six months. 

In our population, we evaluated joint disease activity through the JADAS10 score [7]. 

According with the Declaration of Helsinki, we collected oral and written informed consent from all patients and their cavegivers. The study protocol was approved by the Regional Ethical Committee (RC 03/12).

### 2.2. Clinical and Laboratory Assessment 

Data on demographics, JIA subtype, laboratory tests, clinical manifestations, and therapeutic strategies were collected through medical record reviews. Information about global physical evaluation and complete joint examination available for each visit were also collected. Clinical examination of the ankle was performed by experienced pediatric rheumatologists who evaluated the tibiotalar joint and subtalar joint in order to find any sign of active arthritis [8]. Anterior, lateral, and medial compartment were also evaluated in order to identify tendon involvement [8,9,10]. 

### 2.3. Sonographic Examination

The ankle region was scanned according to guidelines adopted for adults [9]: anterior, perimalleolar medial and lateral scans were obtained for each ankle. Tenosynovitis was diagnosed in the presence of hypoechoic or anechoic thickened tissue detected in 2 perpendicular planes. Fluid within the tendon sheath and Doppler signal may be present as confirmation [8,10].

A pediatric rheumatologist with more than 5 years of experience in rheumatic disease of childhood performed the sonographic evaluation using US machines equipped with high frequency linear probes (>14 Mhz).

## 3. Results

### 3.1. Clinical Results

Sixty-two out of ninety-six patients with JIA developed swelling of one or both ankles during observation time. Forty-eight of them underwent msk-US, and twenty-seven patients presented sonographic signs of tenosynovitis. Median age at diagnosis of JIA was five years (5.4, SD ± 4), and the mean time elapsed between JIA onset and tenosynovitis was 32 months (SD ± 34). Two thirds of patients were females (19 out of 27), and ANA positivity was detected in 16 out of 27 cases. All patients tested negative for rheumatoid factor (RF). Eleven patients met ILAR classification criteria for oligoarticular JIA, seven for polyarticular JIA, and five for oligo-extended JIA. Three patients were classified as psoriatic arthritis, and only one with enthesitis-related arthritis. The median time of follow-up was 4.1 years.

### 3.2. Sonographic Results

A total of 56 swollen ankles of 48 patients were assessed by clinical and sonographic evaluation: 22 ankles showed sonographic signs of joint synovitis (39%), 16 ankles presented signs of both joint synovitis and tenosynovitis (28%), and 14 ankles presented sonographic signs of tenosynovitis only (25%).

Overall, 27 patients (56%) presented a tenosynovitis of one or both ankles during the course of the disease. Beside local swelling, pain and limited range of motion were reported, respectively, on clinical examination of five and eight ankles in patients with tenosynovitis. Twenty out of twenty-seven patients (74%) with tenosynovitis presented an active joint disease (median JADAS−10 score 11.15, IQR 5.12–15.4), but only 14 patients (51%) had a simultaneous ankle arthritis. Tibialis posterior was the most affected tendon, reported on 21 out of 54 ankles, followed by flexor digitorum longus tendon [see Table 1].

### 3.3. Treatment of Patients with Tenosynovitis

Twenty-seven patients received a clinical and sonographic diagnosis of tenosynovitis: 14 out of 27 patients (52%) did not take any medication when tenosynovitis appeared, six patients (22%) were on methotrexate only, four patients (14%) on methotrexate plus biologic agent, and one on biologic agent only. One patient was on anti-inflammatory therapy with NSAIDs at the onset of tenosynovitis, and one with oral glucocorticoids.

In our cohort, 26 patients (96%) experienced the treatment with methotrexate during the course of tendon disease, and 7 out of them (26%) achieved clinical and radiological remission of tenosynovitis without the addition of other therapies. Seventeen out of twenty-seven (63%) were the patients treated with a biological drug for tendon disease: fifteen (88%) achieved tendon disease remission, of which eleven (73%) were in combination therapy with methotrexate. Among responders to combined therapy (methotrexate plus biological), most patients had an oligo-extended or polyarticular form of JIA (respectively, four and three patients), and nine out of eleven had an active joint disease besides tenosynovitis. Patients who achieved tendon disease remission with biological drugs alone were diagnosed with oligo-extended or polyarticular JIA (respectively, one and two patients), and previously failed to respond to methotrexate. Only six patients underwent glucocorticoids tendon sheath injections, and half of them achieved disease tendon remission after the treatment.

Two patients did not achieve tendon disease remission during the study time with any treatment including methotrexate, biological agents, or CS-TSI, and both of them were diagnosed with psoriatic arthritis.

Details of therapeutic management are available in Figure 1.

## 4. Discussion

In our experience, tenosynovitis occurs in more than 50% of patients who presented with ankle swelling, and 50% of these patients did not show ultrasonographic signs of a concomitant joint synovitis in the same region. Overall, 25% of patients with ankle swelling presented isolated tenosynovitis of the foot or ankle. Since swelling is a common clinical sign of both arthritis and tenosynovitis, msk-US has a crucial role to identify tenosynovitis, and recognize the tendons involved. Hendry et al. previously reported a poor agreement between clinical examination and musculoskeletal US in patients with JIA, especially for subclinical disease [11]. Pascoli et al. stated that clinical examination of the ankles in children with JIA is not sufficient on its own to assess all structures affected by inflammation; in their population, the US showed tenosynovitis in 13 ankles out of 31, although clinical examination was normal [5,12].

The presence of a swollen ankle can be interpreted as a sign of arthritis by clinical examination alone, and treated with intra-articular steroid injection with poor results in patients with tenosynovitis. Moreover, a missed diagnosis of tenosynovitis could lead to a wrong JIA diagnosis or JIA-subtype classification [13]. For all of these reasons, msk-US should be routinely used for assessing the presence of joint synovitis or tenosynovitis, and differentiating these two entities in patients with JIA. In recent years, the use of MRI and ultrasonography in pediatric rheumatology has expanded. MRI is the only imaging tool that is able to simultaneously assess all relevant structures in inflammatory joint diseases. However, compared to ultrasonography, MRI must be planned, has longer examination time, higher cost, can value only one target joint at a time, and requires sedation in younger children [14].

To the best of our knowledge, there is limited data about the treatment of tenosynovitis in patients with JIA. Tendon sheath injection with corticosteroids is frequently used to treat tenosynovitis, especially when the ankle region is involved, and some studies reported its effectiveness for disease control [2,15,16]. However, if intra-articular steroid injection is a well-established practice in the treatment of synovitis, the same cannot be said for CS-TSI in tenosynovitis, which requires greater experience of the operator, and shows less encouraging results. Moreover, there is not a consensus treatment for tenosynovitis, nor data on the effectiveness of DMARDs or biological drugs on tendon inflammation in patients with JIA.

In our experience, most of the patients were treated with methotrexate, which was effective only in a small percentage of cases when used as monotherapy. The use of biologic drugs, alone or in association with second DMARDs, allowed the achievement of clinical and ultrasound remission in most cases.

In our experience methotrexate in monotherapy showed appreciable efficacy only in patients with oligoarticular JIA, and in those patients who did not undergo any treatment before.

On the contrary, patients with psoriatic arthritis seemed to present a more severe disease, and required a more aggressive treatment. This would suggest considering psoriatic arthritis in childhood as a separate entity from other forms of JIA, and more similar to the adult disease [17].

Our study had the limitation of being retrospective, with a low sample size, but it is one of the few studies which showed the clinical manifestations of a subgroup of JIA patients with tenosynovitis, and the treatment they underwent. Prospective studies might be needed in order to define the best treatment for JIA patients with tenosynovitis.

## Figures and Tables

**Figure 1 children-09-00509-f001:**
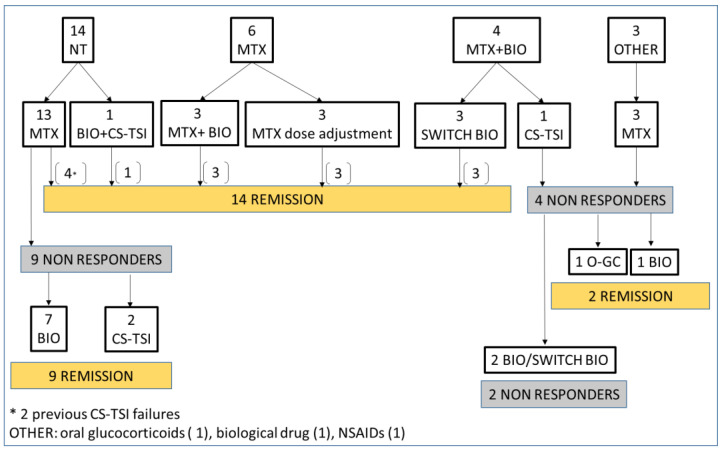
The figure illustrates therapeutic management of patients with JIA since diagnosis of tenosynovitis. NT: no treatment; MTX: methotrexate; BIO: biological drug; CS-TSI: corticosteroids tendon sheath injection; O-GC: oral glucocorticoids.

**Table 1 children-09-00509-t001:** Localization of tenosynovitis.

Tendon	Ankles No. (*n* = 54)	Ankles % (*n* = 54)	Patients No. (*n* = 27)	Patients % (*n* = 27)
Tibialis Posterior	21	38	18	66
Peroneus Longus and Brevis	3	0.5	3	11
Flexor Digitorum Longus	10	18	9	33
Flexor Hallucis Longus	5	0.9	5	18
Extensor Digitorum Longus	6	1.1	6	22
Extensor Hallucis Longus	4	0.7	4	14

## Data Availability

The data that support the findings of this study are available on request from the corresponding author (serena.pastore@burlo.trieste.it).

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
