# Peer review of "Ultrasonographic Assessment for Tenosynovitis in Juvenile Idiopathic Arthritis with Ankle Involvement: Diagnostic and Therapeutic Significance"

_children, 2022, doi:10.3390/children9040509_

Round 1

Reviewer 1 Report

This is an interesting paper on the value of ultrasound assessment for ankle tenosynovitis in children with juvenile idiopathic arthritis.

The authors were able to identify 48 children, of whom a total of 56 ankle joints were scanned, all of which were clinically swollen.

The limitation is the retrospective nature of this work, which is, however, stated in the limitations paragraph. It should be made clear in the abstract and also in the methods that it is a retrospective work.

The exact name of the hospital where the study took place should not be mentioned in the abstract.

A major drawback is the very long recruitment period of 17 years. This raises many questions: How could it be solved organizationally that the same examiner always performed the ultrasound examinations? What influence, if any, did a change in the technical equipment have during this period. How reliable is the clinical documentation of the condition of the ankle joint, since only medical files were evaluated and no study-specific examination was collected at the time of the patient visit?

The abstract should state how often synovitis was detected by ultrasound at the swollen ankle joints. It is mentioned in the results, but it should also be mentioned in the abstract.

What is the value of ultrasound diagnosis of the ankle in children whose joint is not swollen but painful? This would be a very useful addition to the data and would further clarify the need for ultrasound to the reader. 

Are there different approaches to the treatment of synovitis and tenosynovitis? This would underline the significance of the study and should be mentioned in the discussion.

The paper also describes the different therapeutic approaches of the patients, Figure 1. This does not fit the title of the paper at all. Either, the title of the paper is made broader or the information about the therapy of the patients is removed from the manuscript.

Completely missing from the discussion is the possibility of diagnosing tenosynovitis using MRI of the ankle. Reference should be made to MRI literature.

Author Response

This is an interesting paper on the value of ultrasound assessment for ankle tenosynovitis in children with juvenile idiopathic arthritis.The authors were able to identify 48 children, of whom a total of 56 ankle joints were scanned, all of which were clinically swollen. Thanks for your comment. 

The limitation is the retrospective nature of this work, which is, however, stated in the limitations paragraph. It should be made clear in the abstract and also in the methods that it is a retrospective work. We have specified in abstract and in methods that is a retrospective cross-sectional study.

The exact name of the hospital where the study took place should not be mentioned in the abstract. We removed the exact name of the Hospital from the abstract.

A major drawback is the very long recruitment period of 17 years. This raises many questions: How could it be solved organizationally that the same examiner always performed the ultrasound examinations? What influence, if any, did a change in the technical equipment have during this period. As you reported above, we initially enrolled all JIA  patients who developed at least one episode of ankle swelling during a 17-year follow-up period. But only those who underwent msk-us were included into the study; this allowed us to shorten our observation window to about 5 years. During this time the examiner and the technical equipment did not change. How reliable is the clinical documentation of the condition of the ankle joint, since only medical files were evaluated and no study-specific examination was collected at the time of the patient visit? For each patient, we checked medical files of every single visit in order to find if any swelling of the ankle was reported. Since patients with ankle swelling who underwent msk-us were enrolled during the last 5 years of the follow-up period and during this period the examiner was always the same, then we can assume good reliability of clinical documentation.

The abstract should state how often synovitis was detected by ultrasound at the swollen ankle joints. It is mentioned in the results, but it should also be mentioned in the abstract. We added this information even in the abstract.

What is the value of ultrasound diagnosis of the ankle in children whose joint is not swollen but painful? This would be a very useful addition to the data and would further clarify the need for ultrasound to the reader.  We completely agree with the reviewer, it would be really interesting and useful to clarify the role of ultrasound in children with painful but not swollen joints. However, this would be a different study that should start enrolling patients with painful and not swollen ankles. This data is not available in our population.

Are there different approaches to the treatment of synovitis and tenosynovitis? This would underline the significance of the study and should be mentioned in the discussion. We added the following sentence: “if intra-articular steroid injection is a well-established practice in the treatment of synovitis, the same cannot be said for CS-TSI in tenosynovitis which requires greater experience of the operator and shows uncertain results”.

The paper also describes the different therapeutic approaches of the patients, Figure 1. This does not fit the title of the paper at all. Either, the title of the paper is made broader or the information about the therapy of the patients is removed from the manuscript. We have changed the title in “ECOGRAPHIC ASSESSMENT FOR TENOSYNOVITIS IN JUVENILE IDIOPATHIC ARTHRITIS WITH ANKLE INVOLVEMENT: DIAGNOSTIC AND THERAPEUTIC SIGNIFICANCE”

Completely missing from the discussion is the possibility of diagnosing tenosynovitis using MRI of the ankle. Reference should be made to MRI literature. We have added a small paragraph about MRI in the discussion section: “In recent years MRI and ultrasonography use in pediatric rheumatology is expanding. MRI is the only imaging tool that is able to simultaneously assess all relevant structures in inflammatory joint diseases. However, compared to ultrasonography, MRI must be planned, has longer examination time, higher cost, can value only one target joint at the time and requires sedation in younger children.”

(Magni-Manzoni S, Malattia C, Lanni S, Ravelli A. Advances and challenges in imaging in juvenile idiopathic arthritis. Nat Rev Rheumatol. 2012;8(6):329-36.)

Reviewer 2 Report

This study describes the management and outcomes of 27 patients with ankle/foot tenosynovitis by ultrasound, whose clinical exam consisted of joint swelling with or without pain and limited range of motion. The content of this study is quite important, since, as pointed out by the authors, data regarding treatment and response of patients with tenosynovitis in JIA is limited. However several aspects of the manuscript need improvement, as below.

MAJOR COMMENTS:

  • Please clarify retrospective nature of study in methods; this is only mentioned in the discussion briefly.
  • Please explain why the intertarsal joint was considered part of the ankle exam. The study focuses of ankle joint tenosynovitis detection, thus, intertarsal joint swelling should not have been considered as relevant.
  • What was the definition of clinical tenosynovitis? This is unclear in the methods section.
  • How was active disease defined? Validated scores (such as JADAS-10 and others) to define disease activity should have been used – please include this in the description.
  • How was remission defined? Again, validated scores defining remission should be included.
  • Lines 85-93: Median time of follow up is listed twice, and with two different values, please review.
  • The first paragraph of the results is confusing, as the authors do not mention that only 27 patients had tenosynovitis by US until later in the results section, but present proportion of female and ANA positivity results in this first paragraph.
  • Other than ANA, could the authors expand on other JIA markers used for classification and prognosis, including RF, CCP, and HLA-B27?
  • Line 102 in page 3, 20/27 patient with tenosynovitis had active disease: what does this mean? Would expect disease to be considered active in all patients with tenosynovitis; are the authors referring to ankle synovitis being absent in the remaining 7 patients? What was the authors’ definition of active disease?
  • Section 3.2 is overall confusing and difficult to follow. The authors should consider shortening this section as most of this information is included and better explained in Figure 1. The last paragraph of this section is a nice summary which should remain part of the paper.
  • In Figure 1, of those patients without previous therapy who were then treated with methotrexate, 4 are marked as going into clinical remission. Of those, 2 are said to have had previous CS injection failures – at what point were those tried? This is unclear from the figure.

Author Response

Please clarify retrospective nature of study in methods; this is only mentioned in the discussion briefly. We have specified in abstract and in methods that is a retrospective cross-sectional study.

Please explain why the intertarsal joint was considered part of the ankle exam. The study focuses of ankle joint tenosynovitis detection, thus, intertarsal joint swelling should not have been considered as relevant. We completely agree with the reviewer, the presence of the intertarsal joint is confusing, we edited it and removed the intertarsal joint. 

What was the definition of clinical tenosynovitis? This is unclear in the methods section. The diagnosis of tenosynovitis is not made on a clinical basis. The diagnostic suspicion occurs in the presence of swelling in the tendon (medial, lateral or anterior compartment of the ankle) and confirmation occurs with ultrasound.

How was active disease defined? Validated scores (such as JADAS-10 and others) to define disease activity should have been used – please include this in the description. Active disease was defined by JADAS-10. We reported the median JADAS10 score and IQR of patients with active joint disease in the section “Results”.

How was remission defined? Since there are no validated scores to define tenosynovitis remission, we used this term in the absence of clinical and radiological signs of tendon inflammation for at least six months. We added this information in the “Methods” and then we specified “tendon disease remission” every time in the section “Results”.  

Lines 85-93: Median time of follow up is listed twice, and with two different values, please review. We reviewed this part and corrected it. In the old version median time of follow-up  was referred to the whole group of patients with JIA. Now we focus on patients with tenosynovitis. 

The first paragraph of the results is confusing, as the authors do not mention that only 27 patients had tenosynovitis by US until later in the results section, but present proportion of female and ANA positivity results in this first paragraph. We reviewed this part and we mentioned that 27 patients presented sonographic signs of tenosynovitis in the first part of the paragraph.

Other than ANA, could the authors expand on other JIA markers used for classification and prognosis, including RF, CCP, and HLA-B27? In our population all patients with tenosynovitis tested negative for RF and we added this information in the manuscript.  We routinely test for HLA-B27 male patients older than six and patients with enthesitis-related arthritis and in these groups were all negative. We do not routinely test patients with JIA for CCP, so this information is lacking. Since we do not have complete information about CCP and HLA-B27 tests in our population, we did not report them in the manuscript.

Line 102 in page 3, 20/27 patient with tenosynovitis had active disease: what does this mean? Would expect disease to be considered active in all patients with tenosynovitis; are the authors referring to ankle synovitis being absent in the remaining 7 patients? What was the authors’ definition of active disease? We now have specified that 20/27 patients had an active joint disease and even reported median JADAS-10 score.

Section 3.2 is overall confusing and difficult to follow. The authors should consider shortening this section as most of this information is included and better explained in Figure 1. The last paragraph of this section is a nice summary which should remain part of the paper. We edited this part keeping the last paragraph as suggested by the reviewer. 

In Figure 1, of those patients without previous therapy who were then treated with methotrexate, 4 are marked as going into clinical remission. Of those, 2 are said to have had previous CS injection failures – at what point were those tried? This is unclear from the figure. The patients were treated with CS-TSI before the methotrexate and in both cases injections failed. 

Reviewer 3 Report

The study presented, entitled "Ankle involvement in juvenile idiopathic arthritis: don't miss tenosynovitis" by Paolera, describes the sonographic examination of paediatric patients with rheumatoid involvement and the prevalence of tenosynovitis. The authors also discuss the therapeutic response of the respective therapy in the paediatric patients. The study presented clearly and well illustrates the respective study design. For the first time, it shows the prevalence of tenosynovitis in patients with juvenile idiopathic arthritis using musculo-skeletal sonography. This is also correlated with the respective therapy response in this particular patient population. Although the study presented is only retrospective and the population size with 27 patients is not very large, the study presented gives a first good insight into this special field of paediatric rheumatology. 

Author Response

Thanks for your comment and opinion.

Round 2

Reviewer 1 Report

Thank you! 

The article has greatly improved.

I would suggest to change the title to "Ultrasonographic Assessment for Tenosynovitis in Juvenile Idiopathic Arthritis with Ankle Involvement: Diagnostic and Therapeutic Significance"

Author Response

Thanks for your advice. We changed the title as you suggested. 

Reviewer 2 Report

Thank you for addressing the comments

Author Response

Thanks for your comment